# Flavonoids Seen through the Energy Perspective

**DOI:** 10.3390/ijms23010187

**Published:** 2021-12-24

**Authors:** Zhengwen Li, Ming Zhang, Guido R. M. M. Haenen, Lily Vervoort, Mohamed Moalin

**Affiliations:** 1Department of Pharmacology and Toxicology, School of Nutrition and Translational Research in Metabolism (NUTRIM), Faculty of Health, Medicine and Life Sciences, Maastricht University, 6200 MD Maastricht, The Netherlands; zhengwen.li@maastrichtuniversity.nl (Z.L.); l.vervoort@maastrichtuniversity.nl (L.V.); m.moalin@maastrichtuniversity.nl (M.M.); 2College of Food Science and Engineering, Hainan University, 58 Renmin Road, Haikou 570228, China; 3Department of Pharmacology and Toxicology, Cardiovascular Research Institute Maastricht (CARIM), Faculty of Health, Medicine and Life Sciences, Maastricht University, P.O. Box 616, 6200 MD Maastricht, The Netherlands; 4Research Centre Material Sciences, Zuyd University of Applied Science, 6400 AN Heerlen, The Netherlands

**Keywords:** flavonoids, antioxidants, redox modulation, energy perspective, free radical, resilience

## Abstract

In all life forms, opposing forces provide the energy that flows through networks in an organism, which fuels life. In this concept, health is the ability of an organism to maintain the balance between these opposing forces, which creates resilience, and a deranged flow of energy is the basis for diseases. Treatment should focus on adjusting the deranged flow of energy, e.g., by the redox modulating activity of antioxidants. A major group of antioxidants is formed by flavonoids, a group of polyphenolic compounds abundantly present in our diet. The objective here is to review how the redox modulation by flavonoids fits in the various concepts on the mode of action of bioactive compounds, so we can ‘see’ where there is overlap and where the missing links are. Based on this fundament, we should choose our research path aiming to ‘understand’ the redox modulating profile of specific flavonoids, so we can ultimately rationally apply the redox modulating power of flavonoids to improve our health.

## 1. The Dual Nature of Oxygen and Antioxidants

Oxygen is indispensable for us because its reaction with nutrients generates redox energy, which ultimately fuels all our biochemical processes. Remarkably, this ‘burning’ of nutrients is only partly controlled. Part of the liberated energy gives rise to reactive oxygen species (ROS) [1] and reactive nitrogen species [2]. This uncontrolled energy can damage vital biomacromolecules such as DNA, lipids and proteins, which explains how we slowly ‘burn’ during our life [3]. So, oxygen has two faces—good, as well as bad (Yin and Yang). Luckily, there is also a counterforce to the damaging oxidative power of oxygen and other oxidants: the reductive power of antioxidants that can ‘modulate’ the redox power of oxygen and adjust its Yin and Yang.

In a healthy body, the opposing forces of oxidation and reduction are balanced and generate an energy flow that is reasonably controlled, which is essential for life [4]. However, due to a deranged energy flow caused by inflammation, infection, trauma and ‘basal’ metabolic processes, oxidative damage accumulates, resulting in our lifespan being maximized to approximately 120 years [5]. Despite inflicting damage, the deranged energy flow can also switch on redox-regulated transcription factors that act as master switches which turn on the healing process and even adapt us for a second challenge [6]. So, the deranged energy also contains ‘good’. This nicely fits into the dynamic perspective of health; health is not only the absence of disease, but it also includes the ability to adapt to the challenges you have faced [7]. However, there appears to be a limit to this ability to adapt or ‘learning’, as, after multiple hard challenges or a chronic mild challenge, the accumulated damage is too severe to be repaired and the ultimate ‘adaptation’ is death [8]. This is advantageous because ‘death’ prevents a severely damaged cell from thriving and unrestrictedly proliferating to induce cancer. The Yin and Yang of oxygen are illustrated in Figure 1.

The most important antioxidant that modulates the Yin and Yang of oxygen is probably vitamin E, which is directly linked to life. Vitamin E is a dietary factor essential for reproduction. Therefore, it was the named ‘tocopherol’ from the Greek words ‘τόκος’ (birth), and ‘φέρειν’ (carry) with the ending ‘ol’ that refers to the aromatic hydroxyl group that is essential for its antioxidant activity [9,10]. Vitamin E nicely illustrates the link between life and the environment that food provides. A limitation of the food supply with sufficient vitamins will restrict the uncontrolled growth of the population.

For humans, vitamin C is also a dietary antioxidant essential for life. Intriguingly, our inability to make vitamin C is merely due to one nonfunctional enzyme, L-gluconolactone oxidase, the last enzyme in the synthesis of vitamin C [11,12]. It is puzzling what the evolutionary advantage of this mutation is. Interestingly, L-gluconolactone oxidase generates H_2_O_2_ in the synthesis of vitamin C. So, obtaining vitamin C from the diet reduces the endogenous formation of ROS. In evolution, our inability to synthesize vitamin C coincided with our inability to convert uric acid into allantoin. This implies that one antioxidant (vitamin C) is partly replaced by another (uric acid), with another antioxidant profile. The consequence and ‘reason’ for this replacement are still enigmatic.

Another example of redox modulators is the group of flavonoids. Although these compounds do not fulfill the criteria to be labeled as vitamins, they can protect us. A higher intake of these compounds has been associated with a reduced incidence of oxidative stress-related diseases, such as cardiovascular diseases and cancer [13,14,15]. This group comprises over 5000 different compounds [16]. Despite extensive ongoing research on flavonoids for several decades, much about the mode of action of flavonoids is still unknown. We tentatively know that each flavonoid has its own unique ‘biochemical profile’, but these biochemical profiles are only partly unveiled. Moreover, we tentatively know that in different diseases, different types of oxidative stresses are involved, but we still do not know how to choose the flavonoid(s) that best corrects a specific deranged energy flow. Finally, there is no reason that the ‘Yin Yang’ principle of Chinese philosophy, which also forms the basis of Traditional Chinese Medicine (TCM), does not apply to the effect flavonoids have on our health [17]. The dual nature of therapeutic compounds was also acknowledged in the West some 500 years ago by Paracelsus, who stated: “A very good thing worth to be acquired, first has to be separated from the bad. The art is such that nothing good can be acquired without a price. To get what you want, you must face also that you do not want” [18]. The dual nature of antioxidants is already clearly evidenced by various reports on the adverse effects and even toxicity of antioxidants [19].

The objective of the present manuscript is to review how the redox modulation by flavonoids fits into the Western concept of drug action, and the concept of traditional types of medicine. In this way, we can ‘see’ where there is an overlap, and where the missing links are. This provides the fundament we should use to choose our research path aiming to ‘understand’ the redox modulating profile of specific flavonoids, so we can ultimately rationally apply the redox modulating power of flavonoids to improve our health.

## 2. The Mode of Action of Flavonoids Seen through the Current Western Concept of Drug Action

The current concept of drug action, developed in the last century in our Western society, has been very successful in the development of drugs. From the Western perspective, three phases have been identified between the administration of the drug and the final physiological effect. These are: (i) the pharmaceutical phase; (ii) the pharmacokinetic phase; and (iii) the pharmacodynamic phase (Figure 2) [20].

The pharmacodynamic phase describes what the bioactive compound does to the body before finally giving the physiological effect [21], which is the focus of our review. For most drugs, the physiological effect is caused by the interaction of the drug with a large biomolecule, such as a receptor or enzyme. The paradigm coined by Paul Ehrlich is ‘corpora non agit nisi fixata’, i.e., the drug must bind to its bio-target to give its effect. For the bioactivity of compounds, he envisioned that “only such substances can be anchored at a particular part of an organism, as fit into molecules of the recipient complex like a piece of mosaic finds its place in a pattern” [22].

The interaction of a drug with its receptor is subsequently (i) binding the drug to the receptor, (ii) activating the receptor that (iii) triggers a signal cascade of reactions, thus leading to the actual effect (Figure 3). The binding of a drug to the receptor and the dissociation of the drug–receptor complex are governed by the Law of Mass Action [23]. The equilibrium is usually described by K_d_, the apparent dissociation constant of a drug for a receptor.

The binding of the drug to the receptor might activate the receptor. According to the theory of Stephenson, receptor activation produces a certain input stimulus. The magnitude of the input stimulus induced by a certain drug–receptor complex depends on the drug used. Antagonists—in contrast to agonists—do not activate the receptor and consequently do not induce a stimulus when they bind to the receptor. Moreover, the input stimulus generated by the binding of a full agonist is greater compared to that of a partial agonist. Stephenson used the term ‘efficacy’ to express the extent of receptor activation by binding a certain drug to a particular receptor. In the concept of Stephenson, the input stimulus that is produced depends on the efficacy of the drug and the number of receptors occupied [24,25].

Finally, the stimulus generated by the binding of an agonist to the receptor is translated into a physiological response. Often this is a cascade of a biochemical reaction. In each reaction, the input signal is amplified, so that a relatively small number of receptors generates a physiologically relevant effect [26]. In principle, it might be possible to determine how the input signal is amplified in each reaction of the signal transduction cascade. When these functions are combined, it is known how the stimulus is transformed into an effect. However, this approach is too complicated to be practically applicable.

Stephenson tackled this by describing the stimulus-effect transfer as a whole. For a specific receptor in a specific tissue, the stimulus (S) is transferred by an unknown function (f) into an effect (E), in the formula: (1)E=f(S)

Each drug that acts on a specific receptor in a specific organ makes use of the same stimulus–effect function. A high concentration of a certain type of receptor in a tissue, as well as efficient stimulus-effect transduction of this receptor in a specific organ, can direct the effect of a drug that has a high affinity and efficacy for this receptor to this organ [27].

At first sight, the paradigm described above does not seem fit for radical scavenging, redox modulating compounds. To start with, redox modulators do not bind to the energy, they ‘absorb’ the energy. Therefore, the paradigm ‘corpora non agit nisi fixate’ does not seem to fit. Their activity is often expressed by the rate at which antioxidants react with ROS. In principle, this reaction rate is also governed by the Law of Mass Action, but in contrast to the dissociation constant, this is not an equilibrium constant [28]. Another parameter describing the activity of free radical scavenging compounds is the capacity, e.g., the number of ROS that the compound can scavenge [29]. The equivalent for efficacy is the way an antioxidant modulates energy. The way this modulated energy is transformed into an effect involves a complex cascade of biochemical reactions. This looks like the way a drug–receptor complex produces an effect. Therefore, modeling the transformation of redox modulation by antioxidants into a physiological effect probably asks for a similar pragmatic solution to the stimulus-effect function introduced by Stephenson for drugs. In the next few paragraphs, these parameters will be further elaborated.

### 2.1. Radical Scavenging

Flavonoids are very efficient free radical scavengers, better than the endogenous scavengers, vitamin E and vitamin C [30]. In scavenging a free radical, a flavonoid neutralizes the energy of the radical. The flavonoid does so by donating an electron, in addition to or without simultaneously donating a proton, to the radical. The three well-established mechanisms proposed for this are: (i) Hydrogen atom transfer (HAT); (ii) single electron transfer, followed by proton transfer (SET-PT); and (iii) sequential proton loss electron transfer (SPLET). Other mechanisms have also been proposed, such as proton-coupled electron transfer (PCET) and sequential proton loss hydrogen atom transfer (SPLHAT) [31,32,33]. The actual mechanism and efficiency of the scavenging reaction of a specific flavonoid are not only governed by its chemical characteristics but are also dependent on its environment. For example, dissolved in water at a low pH, quercetin hardly displays antioxidant activity, while quercetin dissolved in water at a neutral or higher pH is a strong antioxidant [34]. This suggests that fully protonated quercetin and partially deprotonated quercetin have a different antioxidant profile, probably also with a different mechanism. Moreover, several conformations of the quercetin molecule exist, and each conformer has its unique redox modulating profile [35]. These differences can be predicted and ‘understood’ using quantum calculations.

An important new tool is the quantum chemical calculation, which has evolved tremendously. This can be used to determine where the energy taken up by the radical scavenging is concealed in a flavonoid molecule. Moreover, it can give insight into how the energy flows through the molecule, and how the energy is modulated during this process [36,37]. One of the major restrictions is that it is still not possible to accurately determine the impact of the solvent on the redox behavior of flavonoids.

### 2.2. Capacity, Empowering the Endogenous Antioxidant Network

It should be noted that in scavenging a radical, the antioxidant is consumed as it is oxidized. This means that the capacity of these radical scavenging antioxidants is limited. For example, the antioxidant capacity of vitamin E found in most capacity assays is that one vitamin E molecule can only scavenge two radicals [38]. Flavonoids can scavenge more radicals, e.g., one molecule of quercetin can scavenge up to twelve radicals [39]. Nevertheless, the consumption of the flavonoid during the protection will limit its antioxidant activity.

Luckily, radical scavenging antioxidants do not act in isolation, rather, together they form an intricate network in which the energy of the oxidizing species is captured and transferred from one compound to another [6,40]. For example, in the antioxidant network, oxidized vitamin E can be reduced by vitamin C to form the reduced, active Vitamin E again. In this reaction, Vitamin C is converted into dehydroascorbate. Upon its turn, dehydroascorbate can be reduced to vitamin C by reduced nicotinamide adenine dinucleotide (NADH), a reductant formed during the oxidation of nutrients [41]. Also, most flavonoids can be recycled after oxidation by vitamin C. This interplay between compounds in these networks means that the capacity of flavonoids to absorb energy can also be virtually unlimited.

As flavonoids are very efficient scavengers, they will be the first to pick up the deranged energy. Moreover, we hypothesize that after the flavonoids modulate the energy, it is more efficiently taken up by the endogenous antioxidant network. So, the energy picked up by flavonoids can efficiently be safely shuttled through the antioxidant network to be ultimately neutralized in the metabolic network [42,43]. In this way, flavonoids can empower the protection.

Also, chelation of iron should be considered, as iron is involved in the formation of the most reactive ROS, the hydroxyl radical, in the Fenton reaction. For example, the chelation of iron by the flavonoid monoHER greatly reduces the ability of iron to generate the hydroxyl radical. It has been speculated that this also involves ‘site specific radical scavenging’. Because the hydroxyl radical is formed on the iron atom and the flavonoid is bound to the iron, the flavonoid is at the site where the highly reactive hydroxyl radical is formed. Their close proximity makes it so that the iron-chelating flavonoid can ‘site-specifically’ scavenge the hydroxyl radical at exceptionally high rates that seem to exceed the diffusion rate constant [44].

### 2.3. Empowering the Endogenous Adaptation

As described above, cells contain an antioxidant network that forms a defense system that protects them by absorbing most of the deranged energy flow. When the flow of deranged energy increases, the cell will not only be damaged but will also adapt. The hard energy also turns on a protective sensor, Kelch-like ECH-associated protein 1 (KEAP1), by reacting with the thiol group on the sensor. This will activate the nuclear factor erythroid 2-related factor (Nrf2) pathway, and the cell will make more antioxidants that increase its resilience to the hard energy (Figure 4) [45,46].

By scavenging the radicals, flavonoids take over the hard energy of the ROS and convert it into soft energy. Because the hard energy taken up by the flavonoids flows through the redox modulator, the hard energy becomes ‘soft’. Since this soft energy cannot damage the cell, the cell is protected against a deranged energy flow. Moreover, the soft energy can very efficiently turn on the protection sensor, KEAP1, to finally make more antioxidants [47,48]. With flavonoids, the cell adapts more efficiently to a deranged energy flow by turning on the protection KEAP1 switch without being damaged. In this way, flavonoids can empower the endogenous defense when and where needed.

During cancer therapy, the high energy produced by radiation is aimed to kill tumor cells by turning on their suicide switch, e.g., Glyceraldehyde-3-phosphate dehydrogenase (GAPDH) [49,50]. Because flavonoids will also soften the energy produced by radiation, it is expected that they will turn on the protection switch in tumor cells, as well as prevent the emergency switch from being turned on by radiation [51]. Therefore, taking flavonoids is not advised during cancer therapy. However, in tumor cells even without radiation, the energy flow is already greatly outbalanced. This has already turned on the protection KEAP1 switch, making it one of the reasons that tumor cells are more resistant than healthy cells to chemotherapy and radiotherapy [52]. Moreover, it also implies that the redox modulating effect of flavonoids will be different in tumor cells compared to healthy cells. Interestingly, some findings point out that flavonoids can improve cancer therapy.

In contrast to healthy cells, where energy is primarily generated by mitochondria, tumor cells primary depend on glycolysis for their energy generation, and GAPDH is a pivotal enzyme in this energy generation [53,54,55]. We hypothesize that by focusing the hard energy on GAPDH flavonoids, (i) the emergency GAPDH suicide switch will turn on and (ii) block the major energy supply route. Both effects are specific for tumor cells and will strengthen instead of weakening the antitumor therapy. As outlined above, flavonoids will increase the protection of healthy cells by turning on the protection KEAP1 switch. This will mitigate the deleterious side-effects of antitumor therapy on healthy tissue. At this moment, we are putting this hypothesis to the test.

### 2.4. The Use of the Structure-Activity Relationship (SAR) to Elucidate the Mode of Action

An important strategy to elucidate the mode of action of redox modulators is to establish the relationship between the chemical structure and the effect. Back in 1869, Brown and Fraser already stated that “It is obvious that there must exist a relation between the chemical constitution and the physiological action of a substance”. Intuitively, they put this in a ‘mathematical’ equation:(2)Φ=f(C)

In this equation Φ, the physiological action of a compound depends on the ‘chemical constitution of the compound (which we would now name the chemical properties of the compound) presented by ‘C’, and the ‘constitution’ is converted into a physiological response by an unknown function f. Interestingly, both the vision and equation of Brown and Fraser are identical to the vision and equation of Stephenson. Moreover, both concur with the vision of Ehrlich that physiological active compounds must fit within the molecules of the organism ‘like a piece of mosaic finds its place in a pattern’ [56].

The SAR has gained a pivotal role in studying the physiological activity of compounds. Linking the difference in activity to differences in the chemical structure of compounds gives clues for the interaction of the compounds with the biological system they are interacting with. The groups of both the compound and the biological target that are important for the activity can be identified, as well as the type and the consequence of their interaction, e.g., the change in conformation of the bio-target and the consequence of this [57]. Ultimately, this may lead to the molecular mechanism of the activity of the compounds and an ‘understanding’ of the activity of the compounds. Moreover, the SAR gives clues for the design of more active compounds than the ones tested. In addition, using the SAR can predict that blocking or removing a specific group will make a compound lose its activity. Synthesizing and testing this modified compound can be used to try to falsify and refine the SAR used [58].

In general, a part of the molecule is identical in all compounds that display the same activity via a similar molecular mechanism. This is the ‘pharmacophore’. The pharmacophore is essential for physiological activity, and this is part of the biologically active compounds that fit in the pattern of the mosaic of the organism [59]. Differences in the activities of compounds can be related to the difference in effect that the various substituents have on this ‘fit’. For redox modulators, this includes steric, lipophilic and, especially, electronic effects of the substituent [60].

The use of SAR has been very successful for, e.g., the development of drugs that act on a receptor, enzymes or other large biomolecules [61]. As already outlined above, for radical scavenging, redox modulating compounds, the construction of a SAR is more complex. This is because the ‘biologically active compound’ they react with is much more ‘fluid’ than, for example, the receptor of drugs. This is probably why most of the numerous SAR on redox modulation are descriptive, only stressing the importance of a particular double bond and the presence of substituents at specific places. Another complication is that there might be more than one ‘pharmacophore’ for redox modulation within one molecule, as proposed for quercetin (Figure 5). Moreover, these different pharmacophores probably affect each other and, together, might form a new ‘pharmacophore’ [42,62,63].

In fundamental research, new compounds that do not fit in established relationships are important, as these ‘outliers’ can be used to ‘discover’ a new class of compounds with a new mode of action. Also, well-established SAR should be scrutinized as they might be flawed. For example, even in a series of closely related, simple phenolic compounds, there are indications that their redox modulation activity does not proceed via the same molecular mechanism [64]. Meticulously exploring the activity of relatively ‘simple’ phenolic compounds provides the fundament to finally construct an accurate SAR of the redox modulation of more complex phenolic compounds, such as quercetin.

### 2.5. The Dimension of Time in the Mode of Action

Homeostasis is not a static equilibrium, because energy has to flow to fuel life [65]. This is probably best evidenced by the rise in body temperature when you die. When every reaction in an organism is in perfect equilibrium, the organism is not alive anymore. To be able to live and adapt, there always has to be some energy in the tank.

However, life cannot be modeled as a simple chemical reaction that slowly reaches its equilibrium. It is better modeled as a complex autocatalytic, oscillating reaction in which the driving force slowly fades when we age. This means that not only is time an important dimension, but timing is also relevant. This is because the only constant of life on earth is that our environment was, is and never will be constant. Also, because of this, life has to continuously adapt [66].

The dynamic ‘oscillating equilibrium’, in which compounds are constantly synthesized, modified, broken down and resynthesized, is flexible and versatile. Therefore, oscillating systems seem well-tailored to cope with and adapt to diverse changes. As reviewed previously, several molecular mechanisms that act on a different time-scale are operative to mitigate a deranged energy flow [67]. An instant type of adaptation is the activation of enzyme activity. This is seen for the microsomal Glutathione-transferase 1. Oxidation or alkylation of a thiol ‘sensor’ in the enzyme by ROS or electrophiles can increase the activity of the protective enzyme several times. Interestingly, an excess of ROS or electrophiles inhibits the enzyme. So, by increasing the dose, the response shifts from protection to adaptation, and finally to cell death (Figure 6). When the cell survives, the adaptation can be reversed by the cycle in which proteins are constantly broken down and resynthesized. This all adds to the flexibility of this type of adaptation [68].

Adaptation due to switching on redox-sensitive master switches takes several hours. This type of adaptation is exemplified in Section 2.3, with the transcription factors Nrf2 and GAPDH, which act as a protective switch, and an emergency suicide switch, respectively. As also discussed in that paragraph, redox modulators can be used to turn on these switches in cells. Longer periods of deranged energy can result in the remodeling of organs. In the long run, adaptation can be imprinted on the epigenetic and genomic levels. Also, these types of modulation can be affected by redox modulators. The different types of adaptation discussed do not only differ in their time of onset but their duration and impact are also different, as schematically depicted in Figure 7. Apparently, in modulating an adaptive response, besides the type of the deranged energy flow and the dose and ‘efficacy’ of the redox modulator, the time and timing of the intervention should also be considered [68].

### 2.6. Other Proposed Modes of Action of Flavonoids

As depicted in Figure 8, a myriad of activities of flavonoids has been discovered. It has been speculated that together, these effects—although each might be relatively small—are responsible for the actual health effect. In in vitro experiments, flavonoids can inhibit any enzyme. This is due to the relatively large number of hydroxyl groups present on a flavonoid that can form relatively strong hydrogen bonds with proteins. It has been reported that especially proline residues in proteins are involved in the binding. This ability to bind to proteins is also responsible for the astringent effect of flavonoids [69,70].

The physiological relevance of the direct inhibition of enzymes found in the in vitro experiments is often doubtful, as relatively large concentrations of flavonoids are needed. Also, a great deal of the results found with flavonoids in cell culture experiments needs to be critically evaluated because of the relatively high concentrations used. To date, no receptors for flavonoids have been found. So, it is likely that there is no specific direct effect of flavonoids. Nevertheless, an unspecific, minor inhibition of all enzymes and processes in the cell will slow down the energy spending of all cells and consequently reduce all cell functions, including the formation of ROS. A nice example is the proposed partial inhibition by flavonoids of the poly (ADP-ribose) polymerases (PARP), an enzyme involved in DNA repair. In chronic diseases that induce a lingering PARP-1 (over)activation, the drastically increasing nicotinamide adenine dinucleotide (NAD^+^) turnover results in an increased demand of ATP production for the re-synthesis of NAD^+^. This can provoke an energy crisis in the cells affected. It has been suggested that partial inhibition of PARP-1 by dietary flavonoids can mitigate this energy crisis. Similar, nonspecific inhibitory ‘hormetic’ effects may explain the direct reductive effect flavonoids have on the production of ROS by inflammatory cells, as well as the toxicity of flavonoids [71,72].

Nevertheless, most biological effects can be explained by a redox-modulating effect of flavonoids and not by the binding or direct inhibition of a protein by the flavonoids. As described in Section 2.3, this might involve selectively directing redox energy to redox-controlled master switches. Interestingly, after administration, flavonoids seem to be retained for a relatively long time in a specific part of the body. This is indicative of some sort of specific binding. For example, after IV administration of the flavonoid monoHER to mice, some of the flavonoid seems to be retained in the nucleus of endothelial and muscle cells of the vascular wall, long after all the flavonoid monoHER has been removed from the central circulation [44].

A remarkable observation is that monoHER, in a relatively very low concentration (EC_50_ = 60 nM), is able to protect cells against what seems to be an excess of ROS, namely 200 μM H_2_O_2_. It has been speculated that this is due to three characteristics of monoHER, namely location, location and location: (i) location on the molecular level—monoHER is at the right location for site-specific radical scavenging at a rate that seems to outdo the diffusion rate constant; (ii) location at the supramolecular level—monoHER is located at a pivotal position in the antioxidant network; and (iii) location at the cellular level—monoHER is located in the endothelial and smooth muscle cells in the vascular wall [44].

More research is needed to examine this hypothesis. Interestingly, in this way, the redox modulator monoHER seems to fit in the concept that ‘bioactive substances should be anchored at a particular part of an organism, as fit into molecules of the recipient complex like a piece of mosaic finds its place in a pattern’. Looking through this perspective, the concept of Ehrlich also seems applicable for redox modulators.

## 3. Back to the Future, Connection with Traditional Types of Medicine

To find answers for questions we are faced with, we might look back to find strategies that were used in the past to (re)discover the answer. For example, for the interaction of flavonoids with the antioxidant network, we might learn from one of the fundaments of TCM, namely, the dynamic interaction between herbs. The idea is that carefully selected herbs interact, according to the rule of ‘Jun-Chen-Zuo-Shi’ (Figure 9). Translated into Western terminology, the Emperor (Jun) herb is the compound with the highest efficacy, the Minister (Chen) herb increases the efficacy of the bioactive compounds and reduces their side effect, the Adjuvant (Zuo) herb improves the bio-accessibility and, finally, the Messenger (Shi) herb affects the pharmacokinetics, resulting in targeting [73,74].

All these interaction strategies that date back to the beginning of our era have recently been re-invented for making the effect of Western drugs more specific. Examining the ‘Jun-Chen-Zuo-Shi’ principle in basic pharmacological experiments, such as the response of the combinations in isolated organs or on receptor binding, gives unexpected, contra-intuitive results that open avenues for further research [75,76].

This shows that to connect Western medicine with traditional types of medicine, we should not concentrate on differences but try to find similarities, and skillfully use Occam’s razor. The common denominators in all types of medicine show us where we can find answers. Another common denominator is seen when we look from an energy perspective (Figure 10). In Western medicine, as well as Eastern medicine, opposing forces provide the energy that flows through networks in an organism, which fuels life. In this concept, health is the ability of an organism to maintain the balance between these opposing forces, e.g., homeostasis (West) and harmony (East), which creates resilience. Moreover, strategies used to treat diseases are strikingly alike. Redox modulating compounds and TCM are added to increase or strengthen connections in the network to finally adjust the flow of energy and regain its balance. So, the energy perspective provides a basis to integrate Eastern and Western medicine. Additional research is needed here to connect both worlds [4,77].

Nevertheless, we have to be careful in taking over answers without knowing the right perspective. This can be illustrated using TCM in the West. Western medicine is ‘optimized’ to achieve big therapeutic effects to cure the disease as quickly as possible. Therefore, in the West, only the most effective Emperor herb is used, or even the isolated, most active ingredient in this herb. This, however, gives rise to side effects in the Western world that can be quite severe. To fully appreciate its benefits, and to use its full potential, TCM should be used as developed over centuries in China. In line with the new definition of health, TCM is ‘optimized’ to increase our ability to adapt and to regain ‘harmony’, which has a much more long-term perspective compared to the short-term perspective generally used in the West [77,78,79].

## 4. Conclusions and Future Perspectives

As already stated above, despite a long history of research on the health effects of the redox modulation of flavonoids, much has remained enigmatic. Subjects that need more research are the unexplainable, relatively high-rate constants of the reaction of some flavonoids with radicals, the way flavonoids modulate the energy and how this can empower endogenous protection and the ability to adapt.

To study this, our research strategies have to evolve. Nowadays, research is still conducted in which single antioxidants (e.g., glutathione (GSH)) or, better, the ratio of the reduced and the oxidized antioxidants (e.g., the GSH/ glutathione disulfide (GSSG) ratio) in the blood of patients and healthy volunteers are measured and compared. Although this is valuable, it only gives a narrow, static picture. To illustrate this, the metaphor of examining the effect of water in a house comes to mind—although we are aware that metaphors always give a distorted view. By measuring the GSH levels, one measures the volume of water in the piping system. The GSH/GSSG ratio provides the water pressure in the piping system. A better idea is to observe the way the taps (redox-controlled master switches) are used to control the dynamics of the flow of water through the water pipes (the energy flow through the redox reactions of GSH/GSSG and other redox couples in the networks). It is even better to decipher the mechanism by which water (energy) sustains the lives of the people living in the house, and how it is used to water the plants to make life more beautiful.

Another opportunity is to study other forms of energy, e.g., light. An interesting finding is that TCM ‘corrects’ the light transmitted by the body [80]. This cannot be explained with the Western reductionistic approach yet. A solution might be found in the physiological effect of ultra-weak bioluminescence [81]. Although the molecular mechanism is not well understood, this effect might not only affect the health of an individual, but also the integration and synchronization of members in the group, or even in society. Interestingly, after absorbing the energy of a photon, the redox modulation activity of a flavonoid will drastically change. The impact on the health of ‘excited’ flavonoids is also ‘terra incognita’.

The best way forward is to invent new tools to examine redox modulators. The new perspectives this generates should be connected to the perspectives created using the old tools. To come closer to the solution, the different views should be combined to create a more complete picture (Figure 11).

There are numerous other mysterious ‘forces’ in Eastern medicine and other types of traditional medicine that lack a ‘Western scientific basis’, and are, therefore, left unused. We would like to end with what we consider to be one of the most mysterious forces in the environmental–physiological interaction reported. During and after a war, the shortfall of men caused by war seems to be instantaneously compensated by an increased percentage of boys among newborns. Despite the enigmatic nature of this force that even eclipses that of vitamin E, its universal message is crystal clear: ‘Make love, not war’.

## Figures and Tables

**Figure 1 ijms-23-00187-f001:**
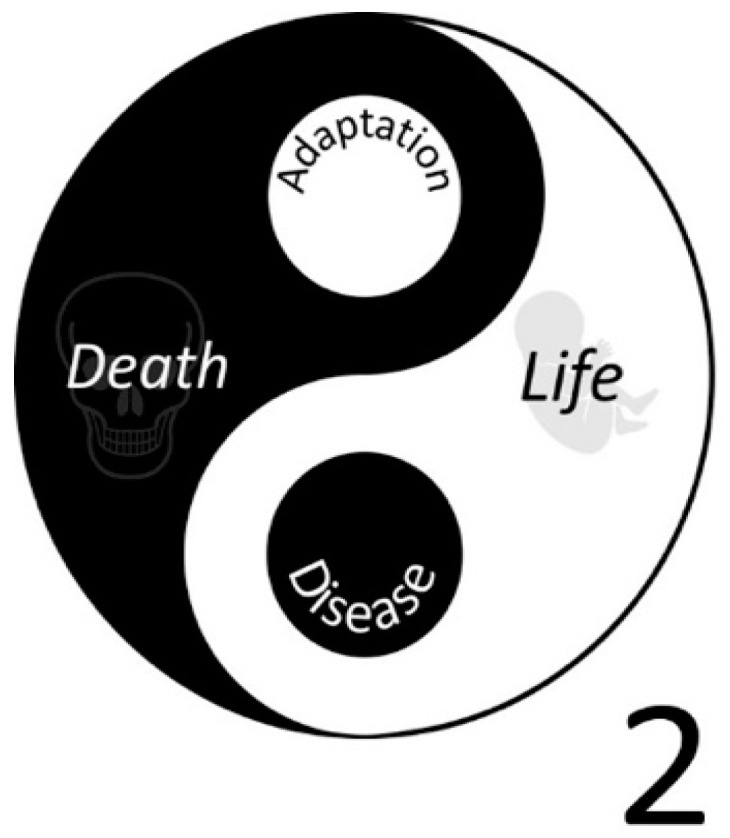
A new symbol for oxygen that reflects its Yin and Yang. In the new symbol, the chemical symbol of Oxygen is merged with the Taiji symbol.

**Figure 2 ijms-23-00187-f002:**
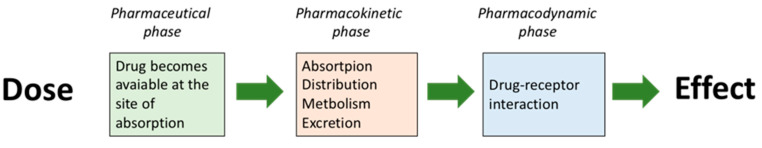
The three phases between the administration of a bioactive compound (drug) and the physiological effect of the bioactive compound.

**Figure 3 ijms-23-00187-f003:**
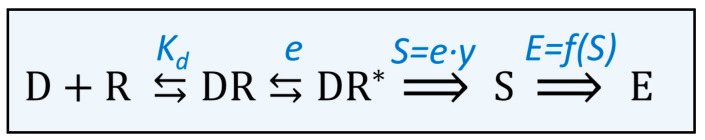
Scheme of the processes in the pharmacodynamic phase of a drug. The drug (D) binds to its receptor (R) and forms a drug–receptor complex (DR). The affinity of the drug for the receptor is described by the dissociation constant (*K_d_*). After binding to the receptor, the receptor can be activated (R*). The potency of a drug to activate the receptor is described by the efficacy (*e*) of the drug. The activated drug–receptor complex generates a stimulus (S) that is proportional to the efficacy of the drug multiplied by the number of receptors occupied (*y*). The stimulus generates an effect (E), which is described by a stimulus effect relationship (*E = f(S)*) that depends on the organ and the type of receptor.

**Figure 4 ijms-23-00187-f004:**
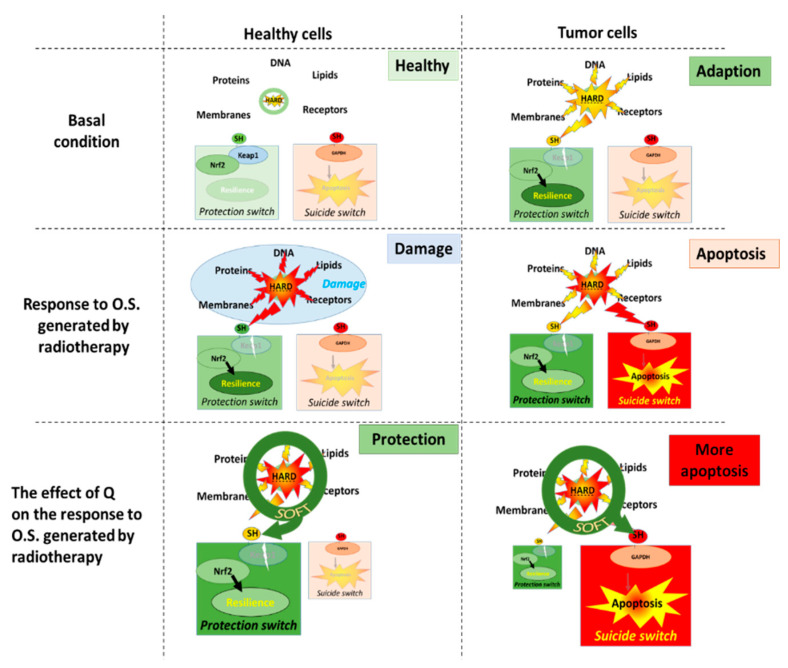
The molecular mechanism that underlies our hypothesis on the difference in the effect of quercetin (Q) on healthy cells and tumor cells. In basal conditions, there is a redox equilibrium between oxidative stress and antioxidant protection. Radiotherapy will damage healthy cells, although the cells will adapt by turning on the protection KEAP1 switch. In healthy cells, Q will prevent damage and increase the protective response, which will increase the resilience of these cells. In tumor cells, the adaptive switch is already turned on in the basal condition. This reduces the efficacy of antitumor therapies which are aimed to induce apoptosis. In tumor cells, Q increases the efficacy of antitumor therapies by turning on the emergency GAPDH suicide switch that induces apoptosis. KEAP1, Kelch-like ECH-associated protein 1; GAPDH, Glyceraldehyde-3-phosphate dehydrogenase.

**Figure 5 ijms-23-00187-f005:**
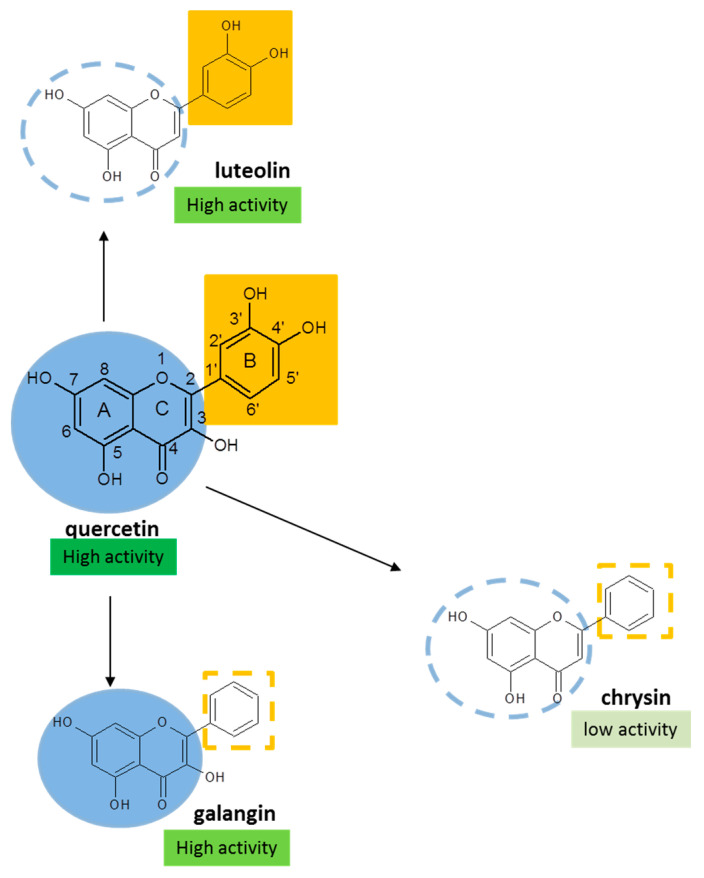
Demonstration of the presence of two antioxidant pharmacophores in Q, the AC-ring (shown in blue) and the B-ring (shown in orange). Q is a very potent antioxidant in the prevention of lipid peroxidation or peroxynitrite scavenging. Galangin, which does not have the 3′OH and 4′OH groups, also has a high activity, demonstrating that the AC-ring of Q is a pharmacophore. Chrysin, without the 3OH group, has a much lower activity than galangin. This shows the importance of the 3OH group in the AC-ring pharmacophore. Luteolin, which does not have the active AC-ring pharmacophore because it misses the 3OH group, is still a very effective antioxidant. This can be ascribed to the B-ring, which is a catechol group, i.e., the other pharmacophore. In Q, which has both pharmacophores, the pharmacophores interact, creating a ‘new’ pharmacophore. The antioxidant activities presented in the figure are taken from Heijnen et al. [42].

**Figure 6 ijms-23-00187-f006:**
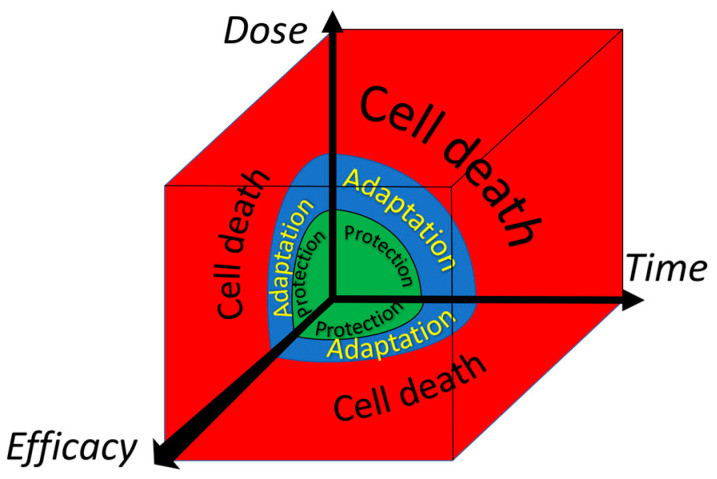
The dependence of the acute adaptive response of the cell to a toxic compound on the dose of the compound, on the efficacy of the compound and on time. Adapted from Sthijns et al. [68].

**Figure 7 ijms-23-00187-f007:**
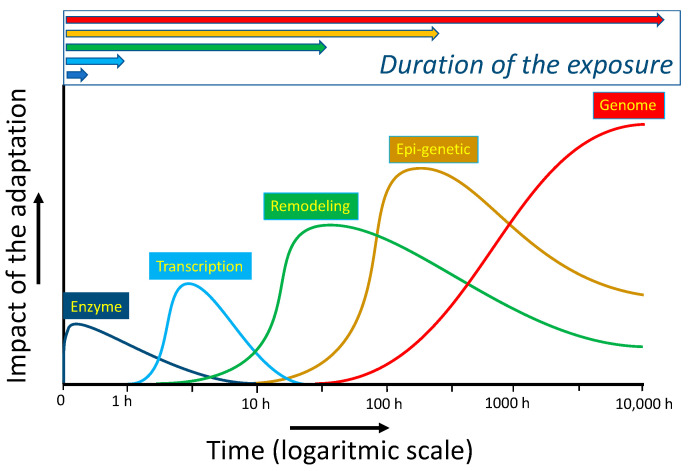
The difference in impact and in the time of onset and duration of several types of adaptations of a cell based on the molecular mechanism of the adaptation. Direct modification of the activity of an enzyme by exposure is an acute adaptive response of cells. In adaptation, due to the activation of the transcription of DNA, there is a lag time in the production of the protective proteins. Remodeling of an organ is a more drastic form of adaptation that takes relatively long and can persist for a relatively long period. Epi-genetic adaptations and especially modifications of the genome can have a large impact, as they can influence the whole population. Adaptation of the level of the genome forms the basis for evolution, which can be seen as an adaptation driven by long-lasting environmental changes, according to the principle of ‘survival of the fittest’. In this paradigm, the genome is, in the end, the ‘information-store’ of the adaption to a persistent change, provided that the organism survives the change. The short time types of adaptation provide the time for the adaptation of the genome to take place. Intriguingly, the genome also stores the information of the other types of adaptation. Adapted from Sthijns et al. [68].

**Figure 8 ijms-23-00187-f008:**
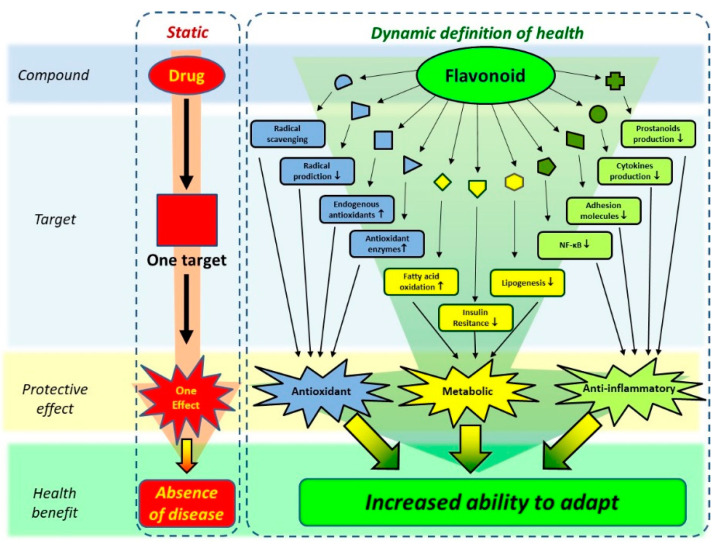
The ‘traditional’ concept of the action of a drug versus the contemporary concept of action of bio-actives such as flavonoids. While traditional drugs are developed to act on a single target, aimed to cure or prevent a disease, flavonoids are thought to act on multiple targets, affecting diverse pathological processes, as well as increasing the ability to adapt. The latter seamlessly fits in the new concept of health, which also includes this dynamic aspect. Adapted from Van de Wier et al. [69].

**Figure 9 ijms-23-00187-f009:**
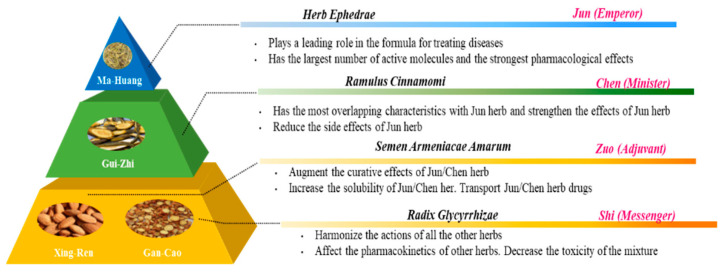
Diagram of the combination principle of Traditional Chinese Medicine (TCM) formula, adapted from Yao et al. [75] and Zhang et al. [76].

**Figure 10 ijms-23-00187-f010:**
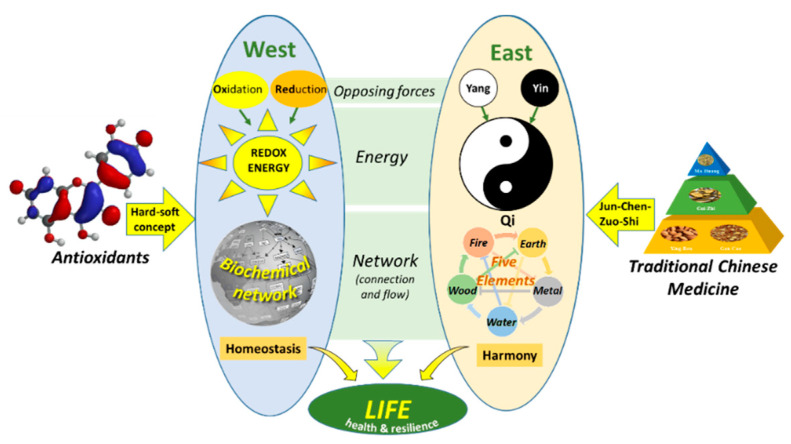
The connection of Western medicine and Eastern medicine from an energy perspective. In both worlds, opposing forces generate the energy that flows through networks, which fuels life. Antioxidants interact with other molecules based on the hard-soft-acid-base concept, which can be used to regain homeostasis. In TCM, different herbs are combined based on the rule of ‘Jun-Chen-Zuo-Shi’ to restore the energy of Qi in the network to regain harmony, taken from Zhang et al. [4].

**Figure 11 ijms-23-00187-f011:**
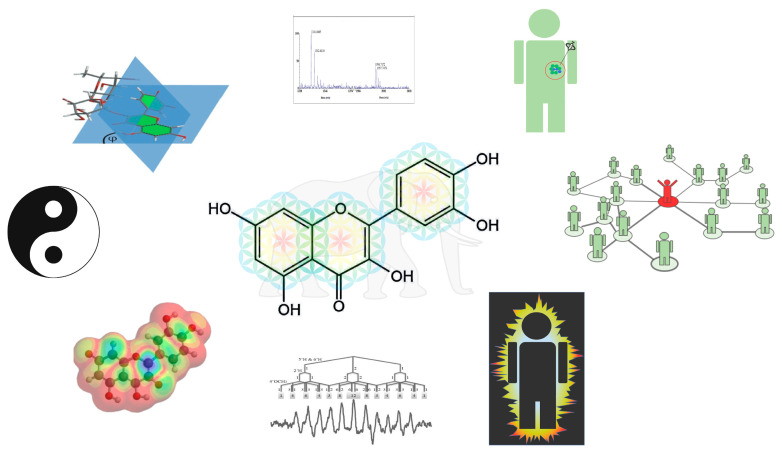
By combining the results obtained with different techniques and from different perspectives, a more complete picture of the redox modulating nature of flavonoids can be obtained. As in the metaphor, one then sees that it is not a tree, a wall, a snake or a rope; it is an elephant. When a closer look is given, one can see through the Yin and Yang and notice its connection to life. The figure is taken from the thesis of Zhengwen Li [36,37].

## Data Availability

Not applicable.

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
