# Peer review of "Flavonoids Seen through the Energy Perspective"

_ijms, 2021, doi:10.3390/ijms23010187_

Round 1

Reviewer 1 Report

Comments to the authors

This is a review covering the redox modulation by flavonoids. Although the review fits one of the scopes of the journal (“fundamental theoretical problems of broad interest in biology, chemistry and medicine”); overall, this review is not well presented nor structured and is not up to data.

Although the manuscript treats an interesting topic that may yield better understanding of mechanisms involved in flavonoids’ mechanisms of action, it is not well focused and it is confusing for the readers.

Title and abstract

- The title is appropriate for the content of the manuscript.

- The abstract should be improved. It is confusing for the readers. The objective of the manuscript is not clear. Do the authors review the antioxidant power of flavonoids or their applications?

Major points

  1. The structure of the manuscript does not have enough quality. For example, in section 1 authors talk about oxygen and its dual nature, but suddenly they talk about their research in modulation of redox processes. Is the manuscript focus on flavonoids or the current research of the authors? The readers will find complicated to follow the article.
  2. In general, the manuscript is very difficult to follow. There is too much information about different topics, and the final goal of the manuscript is missing. The manuscript mixes some general information about antioxidants, the mechanisms of action of flavonoids, the use of SARs strategies… Moreover, point 4.3 “Specific questions that needs further attention” are the future perspectives of the chapter of the doctoral thesis, but I cannot link these questions with all the content of the manuscript.
  3. I encourage the authors to make the manuscript shorter and focus the objective of the review, not including exactly the same content of chapter 1 of the thesis of Z.L. I would include a “Conclusions” section to sum up the highlights of the manuscript.
  4. References are not up to date. The authors should include more recent works. Moreover, a lot of sentences should be accompanied with more references (e.g., when talking about cardio-protection by flavonoids).

Minor points

  1. Some abbreviations and their meanings are not indicated in the text.

- Example: “Reactive nitrogen species” appears for the first time in page 1. The acronym “RNS” has to be indicated. The next times that “reactive nitrogen species” appears; its acronym should be used.

  1. There is an excessive use of words in quotes.
  2. The text has some typos. Please correct.
  3. Figure 1 is not linked in the text. Figures are effective ways of conveying lots of information without cluttering up the text, and they should serve as quick references for the reader. Therefore, they have to be linked to the part of the text they intend to clarify.

Reviewer 2 Report

The work is very interesting and brings many powerful elements of knowledge. All aspects that needed to be clarified were studied and presented accordingly.

Author Response

Please see the attchment. 
